# Transcriptomic profile of the hippocampus of rat strains with contrasting nervous system excitability

**Marina Pavlova, Irina Shalaginova**⬤*, **Natalia Dyuzhikova**

Pavlov Institute of Physiology of the Russian Academy of Sciences, Saint-Petersburg, Russia

* shalaginova_i@mail.ru

## Abstract

Individual variability of reactions to environmental influences which determines the range of the "reaction norm" and the possibilities of adaptation is strongly shaped by inherited properties of the nervous system, including genetically determined differences in excitability. Rat strains selectively bred for contrasting thresholds of neural system excitability provide a model for studying how such inherited differences are reflected at the molecular level. Here, we performed bulk RNA sequencing of the hippocampus in high-excitability (LT) and low-excitability (HT) rats to characterize baseline interstrain transcriptomic divergence. Differential expression analysis revealed strain-specific transcriptional profiles involving not only synapse- and plasticity-related genes, but also non-neuronal components associated with glial/immune functions, intracellular trafficking and protein processing, kinase signaling, extracellular matrix remodeling, and neurovascular regulation. Functional annotation highlighted differences in synaptic organization, neuronal projection development, cellular maintenance pathways, and tissue-level regulatory processes. The analysis motivates testable hypotheses involving synaptic/neurite organization, cellular maintenance pathways (MAPK/PI3K-linked trafficking and redox regulation), and glial/neurovascular components, to be evaluated in follow-up studies using structural and functional tissue-level measurements. This work also provides a reference for cross-model comparisons of polygenic excitability-related traits, as a reference transcriptomic profile from a long-term selective-breeding paradigm.

## Introduction

Individual variability of reactions to environmental influences, including physical, chemical, and psychosocial stimuli, is a fundamental property of organisms with a complex nervous system. One of the main factors determining such variability is the excitability of the nervous system, a genetically encoded trait that is associated with the threshold and dynamics of neural responses [1]. Thus, a deeper understanding of

**Data availability statement:** All relevant data are within the manuscript and its Supporting information files. The raw and processed data in NCBI - GSE327807.

**Funding:** This research was funded by the State funding of the Pavlov Institute of the Physiology, Russian Academy of Sciences (N 1021062411629-7-3.1.4). The funders had no role in study design, data collection and analysis, decision to publish, or preparation of the manuscript.

**Competing interests:** The authors have declared that no competing interests exist.

the molecular and genetic mechanisms underlying inherited differences in excitability is crucial to identify biological risk factors and pathways that determine adaptation reserves and may contribute to vulnerability or resilience to various pathologies, including, but not limited to, stress-related disorders.

One of the effective strategies for investigating complex traits is the use of selectively breeding of model organisms, allowing for controlled genetic divergence along specific behavioral or physiological axes. Notable examples include rats bred for high vs. low anxiety-like behavior (HAB vs. LAB), which differ in coping strategies and HPA axis reactivity [2], and high- vs. low-responder strains (bHR/bLR), which show stable differences in emotional reactivity and PTSD-like behavior [3]. However, these models are typically based on behavioral selection, which may be confounded by uncontrolled factors and the complexity of interpretation of behavioural tests that were used for selection.

In contrast, our model is based on a direct and quantifiable physiological trait – tibial nerve excitability threshold – which reflects fundamental properties of neuronal responsiveness [4]. Long-term selection for this trait has produced two strains: low-threshold (LT, high-excitability) and high-threshold (HT, low-excitability) rats. These strains exhibit stable differences in CNS excitability and behavioral profiles [5,6], making them a valuable system for dissecting the molecular basis of inherited excitability.

Given the hippocampus's key role in regulating neural excitability, emotional processing, and stress response, and its established involvement in excitability-related pathologies [7], we selected this structure to investigate transcriptomic differences between the strains.

The specific tasks were: 1) to determine which genes show differential expression (DEGs) in the hippocampus between LT and HT strains; 2) to characterize the functional structure of the interstrain transcriptomic divergence by Gene Ontology (GO) annotation of DEGs (biological processes, cellular components, and molecular functions); 3) to explore the organization of DEGs into interaction modules by constructing protein–protein interaction (PPI) networks and identifying clustered modules for key enriched functional categories.

This design is not intended to show causal links between gene expression and phenotype, rather, it aims to map interstrain transcriptomic differences in the hippocampus of intact rats formed as a result of selection for high and low thresholds of nervous system excitability. These transcriptional profiles provide a baseline reference for subsequent studies of stress- or aging-related effects in the context of heritable differences in excitability, because they allow to predict the nature of the reaction to stress and other extreme factors depending on the genetic background associated with high and low excitability, as well as to distinguish expression changes that occur along the pre-existing interstrain axes from novel, exposure-induced shifts. In addition, this approach helps to identify a limited set of consistent modules as candidates for subsequent testing of their functional relevance.

## Methods

### Animals

Male rats (5 months old, $n = 5$ per group) from two selectively bred strains (F72)— HT (high-threshold, low-excitability) and LT (low-threshold, high-excitability) were used in this study. The strains were developed at the I.P. Pavlov Institute of Physiology, Russian Academy of Sciences, by selective breeding from an outbred Wistar population, with the aim of studying how neural excitability influences behavioral and physiological phenotypes. These strains are included in the official biocollection at the Pavlov Institute of Physiology, RAS (NoGZ 0134-2018-0003), with patents for selective breeding (No 10769 and 10768).

The selection criterion was the threshold of neuromuscular excitability, determined by electrical stimulation of the tibial nerve (*n. tibialis*) with 2-ms electrical impulses. The first two generations were obtained by sibling mating, and subsequent breeding was performed within each strain. By the 10th generation, the differences between the strains had stabilized, exceeding the within-strain variability by more than fourfold. The animals used in the present study belonged to the 76th generation of this long-term selective breeding program. Nervous system excitability was routinely assessed both during the breeding program, to select animals for further within-line mating, and during the formation of experimental cohorts. Supplementary material (S_1) shows the individual excitability threshold values of the HT and LT rats used in the present study, confirming the phenotypic separation of the two strains in the analyzed cohort.

Animals from both strains were randomly selected for the experiment and kept in a shared vivarium room under controlled conditions (23±2 °C, 12 h light/dark cycle) with free access to standard chow and water. Rats were group-housed (three animals per cage) until 5 months of age, after which they were decapitated in an intact state. Animals were inspected daily by trained animal-care staff, and all procedures were designed to minimize distress. The study involved terminal tissue collection from intact rats at a predefined time point and did not include survival surgery or prolonged invasive manipulations before euthanasia. Decapitation was performed rapidly by experienced personnel to minimize handling time and distress. Anesthesia was not used because agents such as inhalational anaesthetics such as isoflurane can rapidly alter cytokine and other immune-related genes expression [8] which would compromise the primary transcriptomic endpoint of the study and reduce the scientific validity of the terminal sampling procedure. The hippocampus was dissected according to the coordinates of the stereotactic atlas of the rat brain [9]. No humane endpoints were required, as animals were euthanized at the planned time point without prior invasive procedures. All procedures complied with the European Community Council Directive 2010/63/EU and were approved by the Institutional Animal Care and Use Committee of the Pavlov Institute of Physiology, RAS (Protocol No. 09/16, 16.09.2021) and follow the ARRIVE 2.0 guidelines for in vivo experiments. All animals were included in the analysis; no pre-established exclusion criteria were applied.

### Tissue collection, RNA extraction and sequencing

Hippocampal tissue was collected under RNase-free conditions. The brain was rapidly removed and placed on an ice-cooled dissection surface. The whole hippocampus was dissected on ice with care to avoid inclusion of ventricular/ependymal tissue and visible large blood vessels. Tissue was transferred into RNAlater (Thermo Fisher Scientific, Ambion). Total RNA was extracted from hippocampal tissue using Trizol reagent followed by the PureLink RNA Micro Kit (Invitrogen, Carlsbad, CA, USA), according to the manufacturer's protocol after tissue homogenization. Sequencing was performed at the Genoanalitika Lab, Moscow, Russia. The quality and quantity of the isolated total RNA were assessed using a BioAnalyzer with the RNA 6000 Nano Kit (Agilent). The poly(A) RNA fraction was then purified using oligo(dT) magnetic beads (Dynabeads® mRNA Purification Kit, Ambion) according to the manufacturer's instructions.

cDNA libraries for high-throughput sequencing were prepared from the poly(A) RNA using the NEBNext® Ultra™ II RNA Library Prep Kit (New England Biolabs). Library concentrations were measured with the Qubit dsDNA HS Assay Kit (Thermo Fisher Scientific) on a Qubit 2.0 fluorometer, and fragment size distribution was evaluated using the Agilent High

Sensitivity DNA Kit (Agilent, CA, USA). Single-end sequencing was performed on an Illumina HiSeq1500 platform (Illumina, San Diego, CA, USA), generating at least 20 million 50-bp reads per sample. RNA extraction, library preparation, and sequencing were performed simultaneously for all hippocampal samples

Read mapping and gene-level quantification were carried out with *STAR 2.7.9a* [10]. Default settings were applied except for *outFilterMismatchNmax = 3*, which restricted the number of mismatches per read to three. Reads were aligned to the Rnor_6.0 reference genome using *Ensembl v.99* annotation. Raw gene-level counts were generated within *STAR* [10] with the *quantMode GeneCounts* option.

Differential expression analysis was performed in R with *DESeq2 (v.1.28.1)* [11]. Size-factor normalization was carried out with the *estimateSizeFactors()* function based on the median-of-ratios approach, and dispersion parameters were estimated using *estimateDispersions().* Differentially expressed genes were identified with the *DESeq()* function. Group comparisons were evaluated with the Wald test, and p values were adjusted for multiple testing by the Benjamini-Hochberg false discovery rate (FDR) procedure.

## RNA sequencing analyses

DEGs were identified using an FDR-adjusted significance threshold of padj < 0.05. This criterion was used as the primary statistical threshold for differential expression and controls the expected proportion of false discoveries among the genes declared significant. No mandatory log2 fold-change cutoff was applied for DEG definition, because the comparison involved intact animals under baseline physiological conditions, where moderate but statistically consistent expression differences may be biologically meaningful. The additional threshold of |log2FoldChange| ≥ 0.379, corresponding approximately to a 1.3-fold change, was used only to reduce the number of genes displayed in the circular heatmap and to improve visualization of genes with more pronounced inter-line differences.

Gene Ontology (GO) enrichment analysis was performed using DAVID [12] (Database for Annotation, Visualization and Integrated Discovery; DAVID Knowledgebase v2025_1, released April 17, 2025) to identify functional categories overrepresented among the DEGs. Up- and down-regulated genes were analyzed together as a single DEG list, because the aim was to characterize the functional composition of genes differing between the lines rather than to infer direction-specific pathway activation. Gene identifiers were analyzed against the full set of expressed genes detected in our RNA-seq dataset, which was used as the background gene set. GO Biological Process (BP), Cellular Component (CC), and Molecular Function (MF) categories were analyzed. Enrichment significance was assessed using the DAVID functional annotation enrichment procedure, with multiple-testing correction based on the false discovery rate (FDR). Terms were considered significant at p < 0.05 and FDR q < 0.1.

Statistically significant GO terms were manually reduced to a representative non-redundant subset by grouping terms with similar biological meaning and/or overlapping DEG composition and retaining one representative term per group, prioritizing higher specificity, lower FDR values, and relevance to the biological focus of the study.

Data visualization and plotting were performed using the SRplot online platform (https://www.bioinformatics.com.cn; accessed May 1, 2025; updated April 2025) [13]. The following types of plots were generated:

- Volcano plot – selection criterion: *padj* < 0.05 (FDR);

- Principal Component Analysis (PCA) – selection criterion: *padj* < 0.05 (FDR); To avoid circularity associated with PCA based on DEGs, PCA was additionally performed using the transcriptome-wide background expression matrix without DEG-based gene selection. For this analysis, all expressed genes were retained after removal of genes with zero expression or zero variance across samples.

- Circular cluster heatmap – selection criteria: *padj* < 0.05 (FDR) and $|\log_2 FC| > 0.379$; this threshold corresponds to a change in the level of gene expression by more than 1.3 times (or less than $1/1.3 \approx 0.77$ times). The use of an additional filter made it possible to select DEGs with the most pronounced differences in expression levels in order to reduce their

total number to the optimal one for clearer visualization of clusters on a Circular cluster heatmap; clustering based on Euclidean distance, expression data normalized by *z*-score;

- GO/ Pathway Enrichment plots – selection criteria for DEGs is *padj* < 0.05 (FDR); for GO-Enrichment pathways is *p value* < 0.05; *padj* < 0.1.

- The STRING database (version 12.0) was used to construct and analyze the protein–protein interaction (PPI) network. The analysis included both direct (physical) and indirect (functional) interactions derived from curated databases, experimental evidence, and computational predictions. The confidence score threshold was initially set at 0.4 and then increased to 0.7 to retain only high-confidence associations and obtain clearer clustering. Clusters within the PPI network were identified using the Markov Cluster Algorithm (MCL) with the inflation parameter set to 1.5 [14]. For the construction of PPI networks, we focused on differentially expressed genes (DEGs) associated with specific BP term.

Raw and processed RNA-seq data were deposited in the NCBI Gene Expression Omnibus (GEO) under accession number GSE327807.

## Results

### Differential gene expression in the hippocampus of rat strains with contrast levels of neural excitability

The direction and fold changes were determined for the LT strain relative to the HT strain. A total of 654 differentially expressed genes (DEGs) were identified, of which 270 showed higher expression in the LT and 384 in the HT (*padj* < 0.05) (Fig 1).

Inspection of the top DEGs suggested several broad functional groupings: a vascular/immune-associated set; a trafficking/protein-homeostasis-related; and a neuronal signaling/structural. Table 1 presents the top-DEGs highlighted on the Fig 1.

Principal component analysis (PCA) based on the 654 DEGs demonstrated complete separation between LT and HT samples, with the first two principal components explaining 81.7% of the total variance (Fig 2A). The additional PCA was performed using the transcriptome-wide background expression matrix without DEG-based gene selection (Fig 2B). In this analysis, separation between LT and HT samples was also preserved, although the proportion of variance explained by the first two principal components was lower (32.7%), consistent with the use of an unbiased transcriptome-wide gene set rather than genes selected for differential expression.

Fig 3 presents a circular heatmap of the expression profiles of 167 DEGs. The diagram shows two main gene clusters: (1) genes with higher expression in the LT strain (70 DEGs) and (2) genes with lower expression in the LT strain (97 DEGs).

Together, the DEGs, PCA, and heatmap results supported the presence of a structured interstrain expression signature in the hippocampus and motivated subsequent functional annotation analysis.

### Gene ontology (GO) enrichment analysis

To explore the biological significance of the 654 differentially expressed genes, GO enrichment analysis was performed (Fig 4).

Among the identified GO biological process (BP) pathways (Fig 4A), the most enriched and biologically relevant include "*regulation of the MAPK cascade*", "*neuron projection morphogenesis*", "*regulation of neuron projection development*", "*supramolecular fiber organization*" and "*gliogenesis*". Also, the pathways directly related to the regulation of behavior under normal and pathological conditions, such as "*synapse organization*", "*cognition*", "*learning or memory*", "*regulation of synaptic plasticity*", "*generation of precursor metabolites and energy*", and "*extracellular matrix organization*".

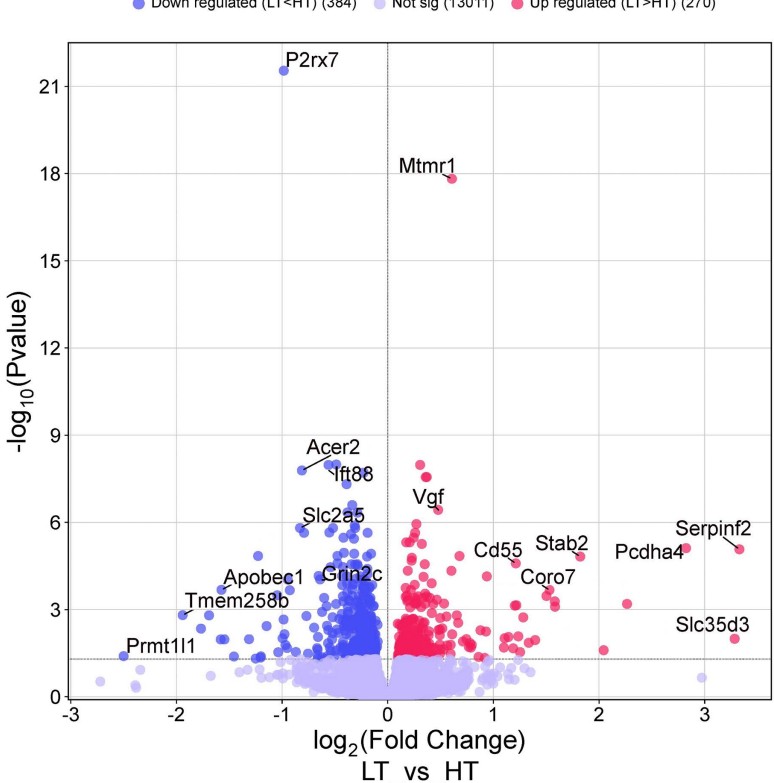

**Fig 1. Differentially expressed genes in the hippocampus of high-excitability (LT) and low-excitability (HT) strains.** Cut off: padj ≤0,05 (FDR).

The most enriched GO pathways for cellular components (CC) (Fig 4B) were "*neuron-to-neuron synapse*", "*side of membrane*", "*extracellular matrix*", "*cell leading edge*", "*distal axon*", "*Golgi apparatus subcompartment*" and "*secretory granule*".

For molecular functions (MF) (Fig 4C), the most significant categories included "*signaling receptor activity*", "*oxidoreductase activity*" and "*calcium ion binding*".

Overall, the GO analysis indicated several major functional trends in the strain hippocampal transcription profiles. Beyond broad synaptic/plasticity-related categories, the enriched terms also pointed to neuronal process development, glial-associated functions, extracellular matrix organization, kinase-related signaling, intracellular trafficking, and metabolic/redox-related processes. Because GO analysis indicated several coherent functional themes, we next asked whether selected categories were also supported by network-level interactions among their gene products.

### Protein-protein interactions

STRING analysis of the complete DEG set (654 genes, FDR < 0.05) demonstrated significant protein–protein interaction enrichment (639 nodes, 547 edges; 381 expected edges; average node degree: 1.71; PPI enrichment $p = 7.77 \times 10^{-16}$), indicating that the transcriptomic differences between HT and LT strains are not randomly distributed across independent genes but are partially organized into interconnected molecular systems (S2 Fig).

To further examine the functional organization of selected DEG subsets associated with biologically relevant GO categories, we next constructed focused PPI networks for genes involved in regulation of synaptic plasticity and regulation of the MAPK cascade (Figs 5 and 6).

**Table 1. TOP-DEGs: Functional roles and general information*.**

| Gene symbol | Gene name | Function | Preferential localization | Higher expression | Log2Fold Change | Padj |
|---|---|---|---|---|---|---|
| Serpinf2 | serpin family F member 2 | Plasmin fibrinolysis inhibition | Extracellular matrix | LT | 3,33 | 8,5E-06 |
| Slc35d3 | solute carrier family 35, member D3 | UDP-glucose transmembrane transporter. Regulating of dopamine signaling. | Golgi apparatus Endosome | LT | 3,28 | 9,9E-03 |
| Pcdha4 | protocadherin alpha 4 | Cell adhesion. Specific neuronal connections in the brain. | Plasma membrane | LT | 2,82 | 7,7E-06 |
| Stab2 | stabilin 2 | May function in angiogenesis, lymphocyte homing, cell adhesion, or receptor scavenging | Cytosol Plasma membrane Macrophages/glia | LT | 1,82 | 1,5E-05 |
| Coro7 | coronin 7 | F-actin regulation | Golgi apparatus Cytozol | LT | 1,53 | 2,1E-04 |
| Cd55 | CD55 molecule (Cromer blood group) | Complement cascade inhibition | Extracellular matrix Plasma membrane | LT | 1,21 | 2,6E-05 |
| Mtmr1 | myotubularin related protein 1 | Predicted to be involved in signal transduction | Cytosol Plasma membrane | LT | 0,61 | 1,5E-18 |
| Vgf | VGF nerve growth factor inducible | Regulation of neuronal activity. neurogenesis and neuroplasticity. | Extracellular matrix Golgi apparatus | LT | 0,48 | 3,7E-07 |
| Prmt1l1 | protein arginine methyltransferase 1 like 1 | Epigenetic regulation and transcriptional control.via histone methylation | Cytosol Nucleus | HT | −2,50 | 4,0E-02 |
| Tmem258b | transmembrane protein 258 B | Oligosaccharyltransferase complex binding activity | Endoplasmic reticulum Plasma membrane | HT | −1,94 | 1,5E-03 |
| Apobec1 | apolipoprotein B mRNA editing enzyme catalytic subunit 1 | m-RNA editing Epigenetic regulation of gene expression | Nucleus Cytozol | HT | −1,58 | 2,1E-04 |
| P2rx7 | purinergic receptor P2X 7 | Purinergic signaling/ Ligand-gated ion channel and is responsible for ATP -possible mechanism of gene expression changes. | Plasma membrane. Mitochondrion | HT | −0,98 | 2,9E-22 |
| Slc2a5 | solute carrier family 2 member 5 | Fructose transporter | Plasma membrane Extracellular matrix | HT | −0,83 | 1,6E-06 |
| Acer2 | alkaline ceramidase 2 | Ceramide metabolism | Golgi apparatus Plasma membrane | HT | −0,81 | 1,6E-08 |
| Grin2c | glutamate ionotropic receptor NMDA type subunit 2C | Encoding of subunit of the N-methyl-D-aspartate (NMDA) receptor | Plasma membrane Endoplasmic reticulum | HT | −0,64 | 9,2E-05 |
| Ift88 | intraflagellar transport 88 | Intraflagellar transport. Positively regulates primary cilium biogenesis. Involved in autophagy. | Cytosol Cytoskeleton Nucleus | HT | −0,56 | 1,1E-08 |

*From DAVID and GeneCards.

To explore functional relationships among the genes contributing to the enriched GO term *"regulation of synaptic plasticity"* (GO:0048167), we constructed a PPI network using the STRING database. The network included 22 nodes connected by experimentally validated or predicted interactions. MCL clustering revealed three functional modules (Fig 5). The network of 29 nodes connected by 51 edges, whereas only 6 edges would be expected for a random set of this size. The network showed highly significant PPI enrichment ($p < 1.0 \times 10^{-16}$), with an average node degree of 3.52 and a local clustering coefficient of 0.497, supporting the presence of a non-random and moderately clustered interaction structure. Cluster 1 (red, 18 genes) represents the core synaptic plasticity module, encompassing key regulators of neurotransmission and cytoskeletal organization such as *Snap25*, *Camk2a*, *Ntrk2*, *Mapt*, and *Gfap*. Cluster 2 (green, 2 genes: *Grin2c*, *Grik1*) corresponds to ionotropic glutamate receptor activity, directly linking to postsynaptic excitatory signaling. Cluster

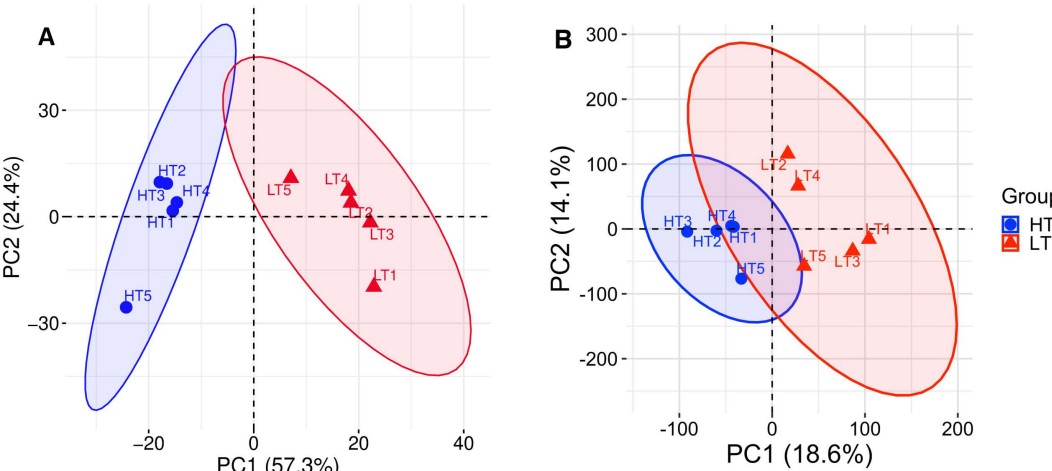

**Fig 2. Principal component analysis (PCA) of hippocampal transcriptomic profiles in HT and LT rat strains. (A)** PCA based on 654 differentially expressed genes (DEGs; padj ≤ 0.05). **(B)** PCA based on all expressed genes after removal of genes with zero expression or zero variance across samples. LT – high-excitability; HT – low-excitability.

3 (blue, *Penk* and *Vgf*) includes neuropeptide-encoding genes associated with neurotrophic signaling and plasticity modulation.

The dense interconnections within Cluster 1 highlight a coordinated regulatory network that integrates synaptic vesicle trafficking, receptor signaling, and structural remodeling processes.

PPI analysis of the functional coherence of differentially expressed genes associated with the GO:0043408~regulation of MAPK cascade pathway using the STRING database revealed five distinct clusters (Fig 6). The STRING network contained 57 nodes and 136 edges, whereas only 33 edges were expected by chance. The network showed highly significant PPI enrichment ($p < 1.0 \times 10^{-16}$), with an average node degree of 4.77 and a local clustering coefficient of 0.476, indicating a non-random and moderately clustered interaction structure.

Cluster 1 corresponds to regulation of phospholipase activity and includes key upstream modulators such as *Agt, Ednra, Pdgfrb, Flt1*, and *Agtr1*. Cluster 2 reflects genes involved in the positive regulation of myeloid leukocyte differentiation. Cluster 3 comprises receptors involved in neurotrophin and GPCR signaling (*Ntrk2, Ntsr2, Gpr37l1, Rit2*). Cluster 4 represents cytoskeletal components involved in microvillus assembly and myosin II binding (*Ezr, Sln*). Cluster 5 includes transcriptional regulators of early cell fate decisions (*Sox2, Bmp4*).

Overall, the PPI analysis helped to resolve selected DEG-associated categories into connected modules. The synaptic plasticity-related network contained modules centered on core synaptic genes, glutamate receptor-related components, and neuropeptide-associated genes, whereas the MAPK-related network was organized into smaller signaling, cytoskeletal, and transcription-related modules.

## Discussion

Top DEGs (Table 1) can be used as concrete candidates for follow-up validation and for building minimal marker panels, which we then relate to the broader GO/PPI structure described below. Several top DEGs may be consistent with a non-neuronal vascular/immune-associated component of interstrain divergence: *Serpinf2* (α2-antiplasmin; coagulation-related) [15], with broader serpin-family links to neurodegenerative conditions [16] and our prior observation of higher *Serpinf2* mRNA in LT amygdala [17]; *Stab2* (endothelial/myeloid transmembrane receptor), consistent with a heightened neuroimmune profile previously reported in LT amygdala [17]; *Cd55* (complement regulation) [18,19]; and glia/

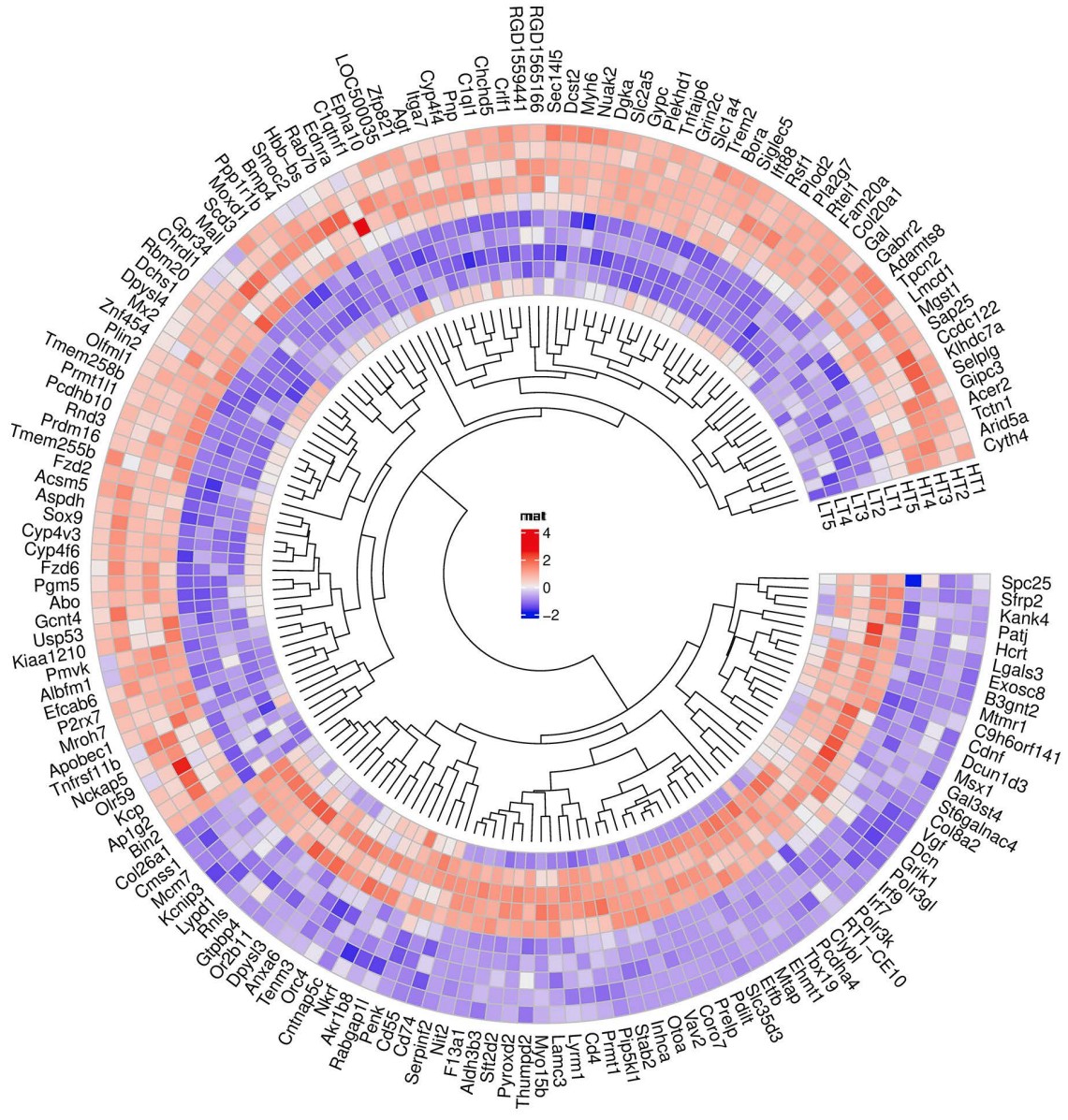

**Fig 3. Cirkular claster heatmap for 167 DEGs expression in the hippocampus of HT and LT rat strains.** Cut off: padj ≤0,05 (FDR), |log2fold change| ≥ 0,379). LT – high-excitability; HT – low-excitability. LT1-LT5, HT1-HT5 – samples; red color – up regulation, blue color – down regulation (LT vs HT).

innate-immunity candidates such as *P2rx7* (predominantly glial purinergic receptor) [20–22] with context-dependent signaling states [23], *Slc2a5*/GLUT5 (microglia-enriched fructose transporter) [24–26], and *Apobec1* (RNA-editing enzyme with reported roles in microglial inflammatory control and neurodegeneration) [27,28] and an epilepsy-related association [29]. *Acer2* (sphingolipid metabolism) [30,31] may reflect differences in lipid/vascular or blood–brain barrier–related components rather than neuron-intrinsic expression changes. Because the present study is based on bulk hippocampal RNA-seq, these interpretations remain tentative and do not provide direct evidence of cell-type-specific expression changes. Further validation would require single-cell RNA sequencing or deconvolution analysis using suitable reference data.

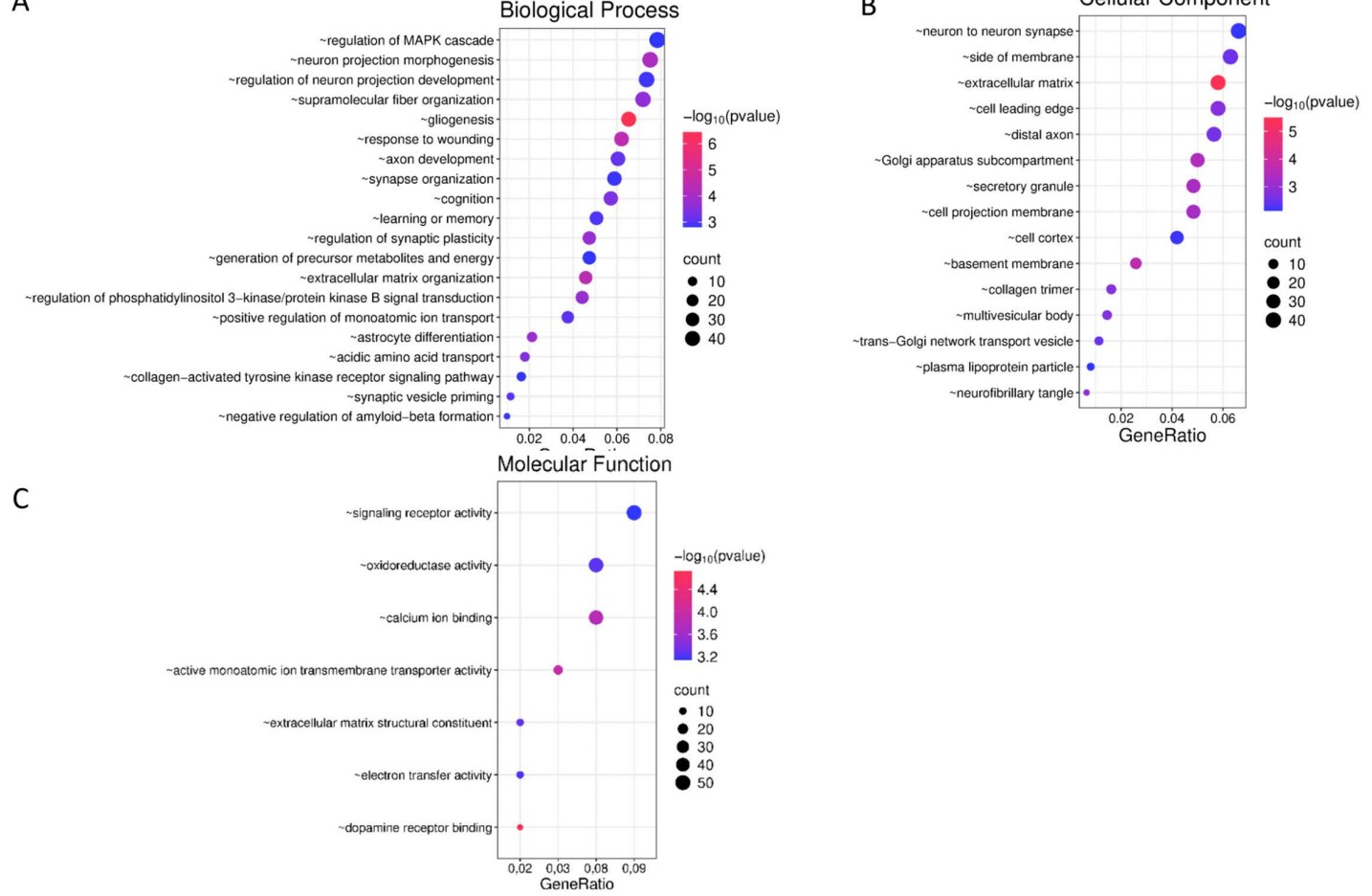

**Fig 4. Gene Ontology-based functional annotation of the differentially expressed genes across biological process (A), cellular component (B), and molecular function (C) categories.** The x-axis indicates the GeneRatio, representing the proportion of DEGs annotated to a given GO term. The color gradient of the dots corresponds to the adjusted p-value (FDR), dot size corresponds to the number of genes annotated to each term.

Several top DEGs form a recurring pattern related to – ER-Golgi/endosomal trafficking and protein homeostasis. *Slc35d3* links to Golgi/endosomal membranes and receptor trafficking pathways (D1 receptor trafficking shown in D1 neurons) [32] and has been connected to trafficking/autophagy-related processes [33]. *Coro7* is a Golgi-localized trafficking/structure factor [34,35]. *Tmem258b* is associated with the oligosaccharyltransferase complex for N-linked glycosylation and ER stress pathways [27]. Together, these genes provide a short list of candidates for follow-up validation in trafficking/ER stress/protein quality control pathways across strains with contrast excitability.

A third group of top-DEGs captures neuronal circuit organization and activity-related signaling/regulation. *Pcdha4* – (PCDHA cluster genes) are linked to neuronal connectivity and synapse specificity [36,37], with protocadherin mutations (e.g., PCDH19) associated with epilepsy and network hyperexcitability [38]. *Vgf* is an activity-regulated neuropeptide precursor induced by neurotrophic factors [39] and associated with hippocampal activity-dependent plasticity and learning [40], with broader biomarker/therapeutic discussions [41]. *Grin2c* (GluN2C NMDA receptor subunit) has distinct receptor properties and cell-type–restricted expression patterns [42], and NMDA receptors are broadly implicated in neuropsychiatric disease contexts [43]. *Ift88* (primary cilia) is linked to neuronal signaling regulation and circuit formation [44–46].

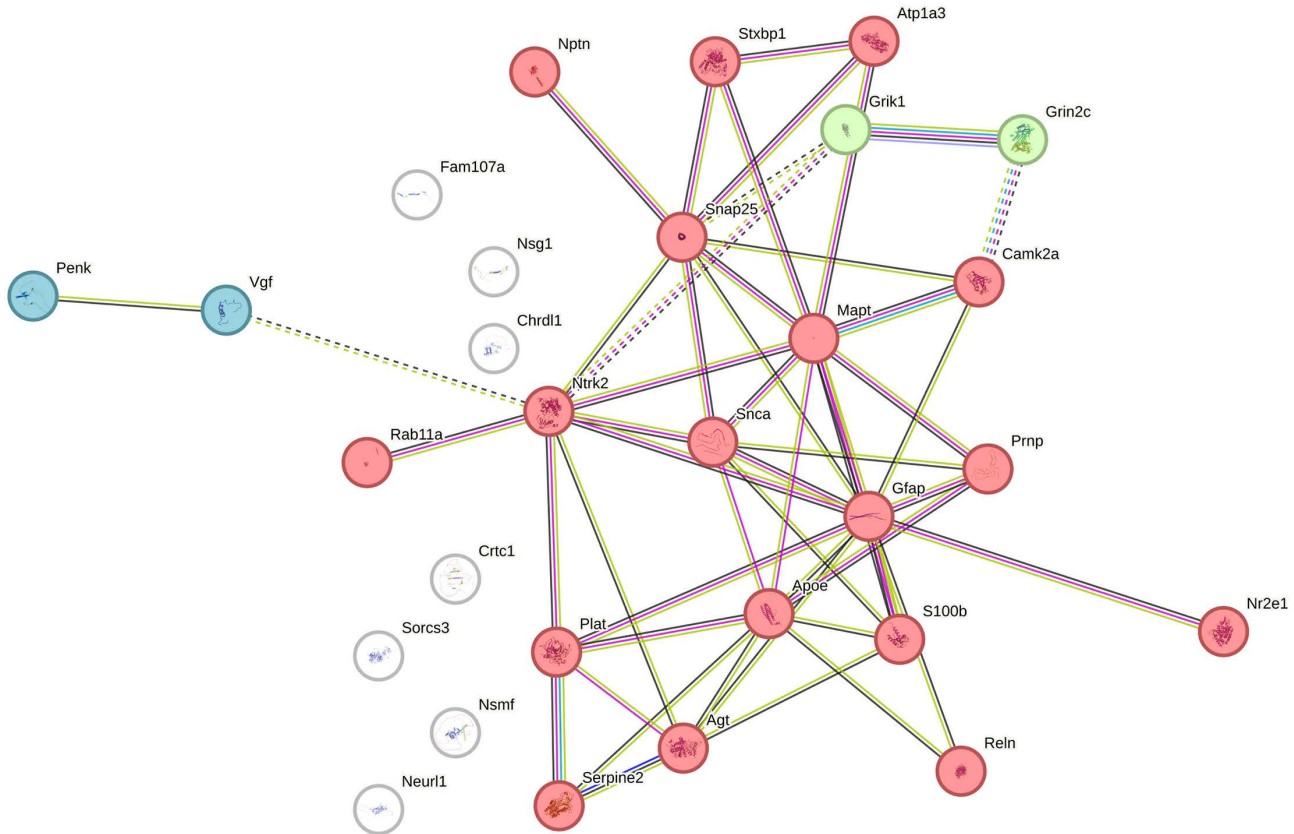

**Fig 5. Protein–protein interaction network for genes involved in regulation of synaptic plasticity.** *Cluster 1* (red, 18 genes) – Regulation of synaptic plasticity. *Cluster 2* (green, 2 genes: *Grin2c, Grik1*) – Ionotropic glutamate receptor activity. *Cluster 3* (blue, *Penk* and *Vgf*) – not annotated in STRING.

*Prmt1l1* (PRMT1-like methyltransferase, poorly characterized) [47], together with PRMT-family links to excitability-related methylation (e.g., NALCN in a PRMT7 context) [48], points to a regulatory layer. Additional candidates in this broader "circuit support" space include *Mtmr1* (myotubularin family; related family links to myelin regulation) [49,50].

Several enriched GO categories were related to synaptic organization and neuronal signaling, suggesting that one component of the baseline interstrain divergence in the hippocampus involves synaptic/plasticity-associated processes. This interpretation is supported by enrichment across BP, CC, and MF categories, although some of these terms are relatively broad (such as "cognition" and "learning and memory") and should be viewed as higher-level functional descriptors rather than pathway-specific evidence on their own. In this context, the hippocampal profile may provide a useful reference for future studies asking whether stress exposure modifies pre-existing synaptic/plasticity-related differences between LT and HT rats or instead introduces new changes outside this axis.

Interestingly, we also found significantly higher expression of *Grin2c* in the HT strain, encoding the GluN2C subunit of the NMDA receptor, which may be relevant to differences in glutamatergic signaling and synaptic modulation. Other model demonstrate that resilient mice exposed to chronic stress show compensatory structural plasticity of postsynaptic compartments in the CA1 hippocampus and a shift of the AMPAR/NMDAR ratio toward NMDAR, whereas animals developing anhedonia display an increased AMPAR/NMDAR ratio [51]. Since AMPARs mediate fast excitatory currents and NMDARs, which require prior depolarization, allow $Ca^{2+}$ influx and trigger long-term plasticity processes such as LTP [52], such shifts

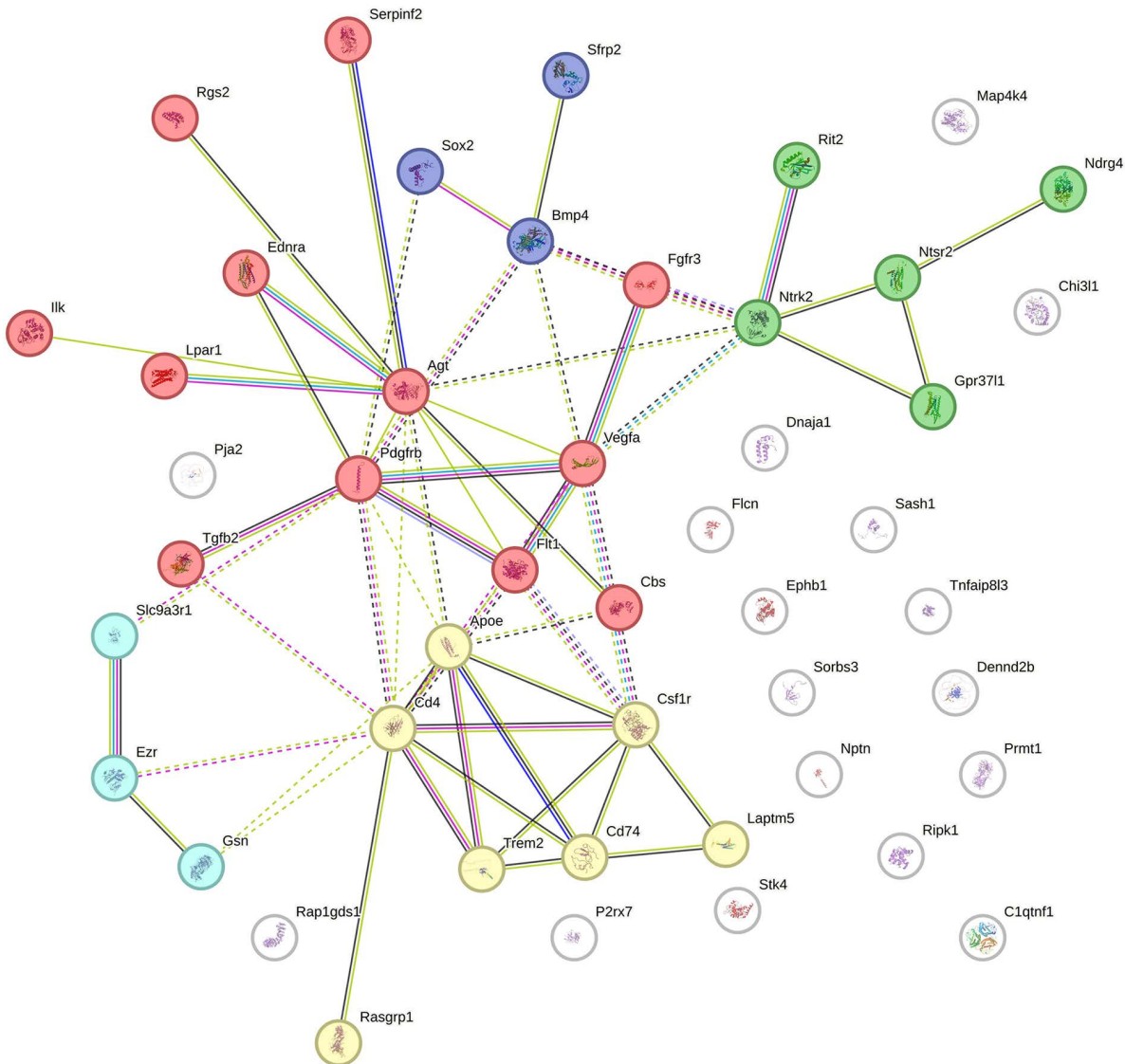

**Fig 6. Protein–protein interaction network for genes involved in regulation of MAPK cascade.** *Cluster 1* (red, 12 genes) – regulation of phospholipase activity, *Cluster 2* (yellow, 7 genes) – positive regulation of myeloid leukocyte differentiation, *Cluster 3* (green, 5 genes) – receptors involved in neurotrophic and GPCR signaling, *Cluster 4* (cyan, 3 genes) – cytoskeletal components involved in microvillus assembly and myosin II binding, *Cluster 5* (purple, 3 genes) – transcriptional regulators of early cell fate decisions. Genes not integrated into any cluster due to insufficient interaction evidence are shown as unconnected white nodes.

in their balance raise the possibility of a molecular mechanism that may contribute to differences in the adaptability of hippocampal circuits associated with resilient or vulnerable behavioral phenotypes.

Another major enrichment theme involved neuronal process development and glial-related terms, including neuron projection morphogenesis, axon development, gliogenesis, and astrocyte differentiation, with corresponding CC enrichment in the cell projection membrane and cell cortex. The cell cortex is a thin, actin-rich layer underneath the plasma membrane that regulates membrane mechanics and shape; it is key to cell migration, division, and tissue morphogenesis, and has been implicated in cell differentiation during development [53]. The cell projection membrane refers to the specialized

plasma membrane of neuronal extensions, such as axons and dendrites, where ion channels and receptors are concentrated [54]. Although our data identify this pathway only at the level of enrichment, its biological significance is well documented. This localization is functionally critical, for instance, the axon initial segment contains a high density of voltage-gated channels that makes it the dominant site for action potential initiation and regulation of synaptic inputs integration, intrinsic excitability, and transmitter release [55]. In hippocampal pyramidal neurons, ion channels such as Kv4.2 and HCN are differentially distributed along dendrites and shape local excitability and input integration [56]. Thus, enrichment of these CC terms and also "active ion transporter activity" in MF in our RNA seq dataset may reflect genetic differences between LT and HT rats in the molecular organization of neuronal processes, formation of dendritic-axonal architecture pointing to basis to variation in excitability and information processing strategies at the circuit level. In practical terms, these enrichment signals suggest that the interstrain divergence may involve differences in neuronal process architecture and compartmentalized membrane organization; therefore, the most informative follow-up study is at the morphological and functional levels (neurite/axon initial segment organization and circuit excitability measures).

Differences in term "gliogenesis" may indicate that the strains have different patterns of glial activation upon stimulus. For example, one possible interpretation is that hippocampal astrocytes may differ numerically or functionally: recent data have shown that hippocampal astrocytes are actively involved in regulating anxiety, responding with calcium waves to disturbing environmental stimuli and influencing behavior [57]. More reactive astrocytes can better control the concentrations of neurotransmitters (glutamate, GABA) and maintain a balance of excitation/inhibition.

A next enrichment clusters stress/adaptive-response terms (e.g., "response to wounding", MAPK and PI3K/AKT regulation) together with Golgi/endosomal compartments (trans-Golgi network, secretory granules, multivesicular bodies) and oxidoreductase activity. While the Golgi apparatus and multivesicular bodies are known to participate in receptor trafficking and thereby contribute to synaptic plasticity under physiological conditions, they also play a crucial role in neuronal defense by regulating protein turnover, receptor degradation, and adaptive membrane remodeling during cellular stress [58]. Notably, in models of systemic inflammation/hepatic encephalopathy, increased multivesicular bodies in astrocytes has been interpreted as a compensatory response supporting gliovascular homeostasis [59], suggesting that astroglial endosomal stress-handling pathways may differ between LT and HT strains at baseline.

Similarly, MAPK and PI3K/AKT signaling pathways are not only central to long-term synaptic plasticity but also constitute major survival and stress-response cascades in neurons [60]. Genetic alterations in the MAP-ERK pathway are associated with neurodevelopmental disorders, while its aberrant activation in glial cells contributes to neuroinflammatory mechanisms of major neurodegenerative diseases such as Alzheimer's, Parkinson's, Huntington's disease [61].

Enrichment of the MF term "oxidoreductase activity" suggests the involvement of antioxidant and redox-regulatory processes that may support neuronal homeostasis under conditions of increased excitability or metabolic demand in the brain [62]. Together with enrichment of Golgi/endosomal and MAPK/PI3K-related categories, this points to a baseline cellular-maintenance component of the interstrain hippocampal divergence. An important next step is to determine whether this signal is linked to specific cellular compartments, such as astrocytes, endothelium, or microglia, and whether it is reflected at the structural or protein level in the hippocampus.

Another enrichment theme was related to extracellular matrix organization. In the hippocampus, ECM, including perineuronal nets, not only provides structural support but also modulates plasticity, memory-related processes, receptor mobility, and the balance between excitation and inhibition [63]. Its role in plasticity regulation and in stress- or depression-related remodeling has been well documented [64].

Enrichment of metabolism-related categories further suggests strain-specific differences in energetic support and neurotransmitter-associated metabolic processes. For example, mitochondrial calcium uptake can enhance electron transport chain activity and ATP production during high-frequency firing in hippocampal neurons [65], while amino acid transporters involved in the glutamate-glutamine cycle help maintain transmitter supply under sustained synaptic load [66].

Thus, changes in gene expression in the hippocampus of LT and HT rats affect processes at several levels, from synaptic transmission and process morphogenesis to regulation of the extracellular environment and metabolic support, which provides the basis for further study of their functional significance.

The significant PPI enrichment of the complete DEG set indicates that the hippocampal differences between HT and LT rats are organized beyond the level of individual differentially expressed genes. The higher-than-expected number of interactions suggests that the DEG set is concentrated within functionally connected molecular neighborhoods. This is consistent with the nature of inherited nervous system excitability as a polygenic trait and selection is unlikely to produce a transcriptional signature restricted to a single canonical pathway. Instead, the enriched interaction structure suggests that baseline hippocampal divergence is distributed across connected molecular neighborhoods involving synaptic organization, signaling, cellular-maintenance, and glial/neurovascular components. This supports the view that the interstrain difference represents a coordinated tissue-level transcriptomic state linked to the long-term selective-breeding history of the strains, and provides the context for the focused analyses of synaptic plasticity- and MAPK-related DEG subsets.

In the synaptic plasticity-related network, several proteins point to processes such as vesicle cycling, calcium signaling, and synapse structural organization. Among them, SNAP25 (node degree = 9, S3 Table) is especially notable because, as a core SNARE-complex protein, it is directly involved in synaptic vesicle fusion and neurotransmitter release. Altered SNAP25 expression has been linked to cognitive and emotional disorders [67], and SNAP-25 also contributes to postsynaptic receptor trafficking and dendritic spine organization, indicating that its variation may influence both pre- and post-synaptic components of plasticity [68]. Another important node of this cluster is NTRK2 (TrkB, node degree = 9) – a BDNF receptor associated with plasticity, synaptic stability, and adaptive response to stress [69].

MAPT (tau, node degree = 10) is now considered not only as a protein "associated with microtubules". There is increasing evidence of its involvement in synaptic regulation, microtubule stabilization, dendritic spike support, and memory consolidation. Tau removal in mice leads to age-related disorders of short-term memory and defects in synaptic plasticity [70]. The role of synaptic tau, i.e., tau localized in synapses, in the coordination of synaptic responses and plasticity is also discussed [71].

GFAP (glial fibrillar acid protein, node degree = 12) is a marker and structural component of astrocytes involved in the morphological plasticity of astrocytic processes and the regulation of homeostasis. Astrocytes and GFAP are important not only for supporting neurons, but also for modulating plasticity (for example, through environmental regulation, glutamate clearance, $Ca2^+$ signals) [72]. Mutant GFAP mice models show decreased synaptic plasticity and impaired astrocyte function [73]. The differences in *Gfap* expression between the strains are consistent with the astrocyte stress response that we found earlier: 7 days after stress, the LT strain showed a decrease in the number of GFAP+ cells in the prefrontal cortex, hippocampus, and amygdala, whereas in HT rats such a decrease was limited only by the hippocampus [6].

The presence of MAPT and GFAP in this cluster indicates that the plastic changes reflected in the transcriptome are not limited to neurotransmitter mechanisms in synaptic transmission, as well as stabilization of the cytoskeleton, changes in astrocytic support. Such nodes are particularly interesting as candidates for subsequent verification: how can interstrain differences in MAPT expression affect the structural stability of dendrites in response to stress, and variations in GFAP expression affect glial reactivity and its ability to "service" neural networks?

Among the genes contributing to the MAPK cascade-related network, angiotensinogen (AGT, node degree = 12, S6 Table) is notable because angiotensinogen is expressed in both astrocytes and neurons [74]. Interstrain differences in basal AGT mRNA levels have been reported previously in rat: in Wistar astrocytes expression was higher compared to the spontaneously hypertensive rats (SHR) strain [75]. AGT is a precursor molecule of Ang II, which in astrocytes causes activation of several intracellular signaling pathways involving mitogen activated protein (MAP) kinases, tyrosine kinases, protein kinase C (PKC), immediate early response genes [76]. FLT1 (VEGFR1, node degree = 5), a receptor for vascular growth factor, is involved in neurovascular plasticity: its expression in the hippocampus increases after ischemic damage, which may reflect an attempt to regenerate or restructure the vascular network [77]. Differences in the expression of AGT

and FLT1 suggest that the interstrain divergence may extend beyond synaptic genes to include pathways related to vascular signaling and possibly glia–vascular interactions within hippocampal tissue.

PDGFRB (platelet growth factor β-form receptor, node degree = 10) is involved in MAPK activation through binding to the growth factor receptor binding protein 2 (GRB2) and triggering the Ras/MAPK cascade. Multiple transcription factors, such as Elk-1, c-Jun and c-Myc, are catalyzed by MAPK activation and control cell metabolism, growth, and proliferation [78]. In nervous tissue, PDGF/PDGFR signaling regulates neurogenesis, cell survival, and remodeling of neural networks [79].

Thus, the largest cluster identified by the PPI analysis of genes associated with GO:0043408 "regulation of the MAPK cascade" included many modulators of kinase signaling pathways potentially relevant to vascular, glial, and hormonal responses.

## Conclusion

The identified hippocampal transcriptomic profile provides a molecular baseline for the interstrain differences that have emerged through selection and helps identify candidate pathways that may contribute to differential responses to stress and other extreme factors depending on the differences in genetic background associated with high and low excitability.

Transcriptomic analysis of the hippocampus in rat strains with genetically determined differences in nervous excitability revealed divergence in molecular profiles not only at the level of neuronal factors related to neurons excitability, synaptic organization and structural neuroplasticity, but also non-neuronal factors related to glial function, cellular stress responses, MAPK signaling, extracellular matrix remodeling, and vascular regulation. This baseline profile provides a reference framework for interpreting future studies in this model, helping to distinguish constitutive interstrain differences from changes induced by experimental manipulations. Although the present bulk RNA-seq approach does not provide direct cell-type resolution and therefore requires further validation by cell-type-resolved methods, it help identify the major functional domains in which interstrain hippocampal divergence is organized. The analysis motivates testable hypotheses involving synaptic/neurite organization, cellular maintenance pathways (MAPK/PI3K-linked trafficking and redox regulation), and glial/neurovascular components, to be evaluated in follow-up studies using structural and functional tissue-level measurements. This work also provides a reference for cross-model comparisons of polygenic excitability-related traits, as a reference transcriptomic profile from a long-term selective-breeding paradigm.

## Supporting information

**S1 Fig. Individual excitability thresholds in selectively bred HT and LT rats.**
(DOCX)

**S2 Fig. STRING PPI network from the full set of DEGs identified between HT and LT hippocampus samples.**
(PNG)

**S1 Table. Gene Lists.**
(XLSX)

**S2 Table. Fig 5 string MCL cluster descriptions.**
(XLS)

**S3 Table. Fig 5 string MCL clusters.**
(XLS)

**S4 Table. Fig 5 string node degrees.**
(XLSX)

**S5 Table. Fig 6 string MCL cluster descriptions.**
(XLS)

**S6 Table. Fig 6 string MCL clusters.**
(XLS)

**S7 Table. Fig 6 string node degrees.**
(XLSX)

**S8 Table. String node degrees of PPI network from the full set of DEGs.**
(TSV)

## Acknowledgments

We would like to thank Genoanalytica Lab, Moscow, Russia, where the samples were sequenced and the initial data processing was carried out. We would also like to express our sincere gratitude to Professor Natalia Kudryavtseva, Doctor of Biological Sciences (Pavlov Institute of Physiology of the Russian Academy of Sciences), Olga Redina, Doctor of Biological Sciences (Institute of Cytology and Genetics of the Siberian Branch of the Russian Academy of Sciences) and Artur Shvetcov, Doctor of Philosophy (Senior Bioinformatician in Westmead Institute for Medical Research, Sydney, Australia) for their assistance and support in working with transcriptome analysis data.

## Author contributions

**Conceptualization:** Marina Pavlova, Irina Shalaginova, Natalia Dyuzhikova.

**Formal analysis:** Marina Pavlova.

**Funding acquisition:** Natalia Dyuzhikova.

**Investigation:** Irina Shalaginova.

**Visualization:** Marina Pavlova.

**Writing – original draft:** Irina Shalaginova.

**Writing – review & editing:** Marina Pavlova, Natalia Dyuzhikova.

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
