## [Decision Letter · Decision Letter 0]

17 Mar 2026

PONE-D-26-04533Transcriptomic profile of the hippocampus of rat strains with contrasting nervous system excitabilityPLOS One

Dear Dr. Shalaginova,

Thank you for submitting your manuscript to PLOS ONE. After careful consideration, we feel that it has merit but does not fully meet PLOS ONE’s publication criteria as it currently stands. Therefore, we invite you to submit a revised version of the manuscript that addresses the points raised during the review process.

We look forward to receiving your revised manuscript.

Kind regards,

Jean-Pierre Mothet, Ph.D

Academic Editor

PLOS One

**Journal Requirements:**

https://doi.org/10.1371/journal.pone.0323325

In your revision ensure you cite all your sources (including your own works), and quote or rephrase any duplicated text outside the methods section. Further consideration is dependent on these concerns being addressed.

4. Please note that funding information should not appear in any section or other areas of your manuscript. We will only publish funding information present in the Funding Statement section of the online submission form. Please remove any funding-related text from the manuscript.

5. Please ensure that you refer to Figure 6 in your text as, if accepted, production will need this reference to link the reader to the figure.

6. Please upload a new copy of Figures 5 and 6 as the detail is not clear. Please follow the link for more information:  https://journals.plos.org/plosone/s/figures

7. We are unable to open your Supporting Information file ( Supplementary_1 string_MCL_2_GO_1.tsv and Supplementary_2 string_MCL_2_GO_6.tsv) Please kindly revise as necessary and re-upload.

**Additional Editor Comment:**

The two reviewers raised several concerns about methodologies that are presented without enough details notably regarding animal welfare, description and justification of the selected model, and for the GO analysis. Furthermore, they found that PPI analysis does not reflect an independent layer of experimental evidence from the present study. Finally, it is not clear why the differential gene expression analysis has been performed with a log2FoldChange filter >0.379 and not >1 as classically used and how it may impact the conclusion. Although, I personally think that your manuscript has some merits, I invite you to address those important issues during revision.

Reviewers' comments:

Reviewer's Responses to Questions

**Comments to the Author**

1. Is the manuscript technically sound, and do the data support the conclusions?

Reviewer #1: Partly

Reviewer #2: Partly

Reviewer #3: Partly

2. Has the statistical analysis been performed appropriately and rigorously? 

Reviewer #1: Yes

Reviewer #2: N/A

Reviewer #3: I Don't Know

3. Have the authors made all data underlying the findings in their manuscript fully available?

Reviewer #1: Yes

Reviewer #2: Yes

Reviewer #3: No

4. Is the manuscript presented in an intelligible fashion and written in standard English?

Reviewer #1: Yes

Reviewer #2: Yes

Reviewer #3: Yes

5. Review Comments to the Author

Reviewer #1: The presented manuscript aims to study the inherited properties of the nervous system to respond to external environmental cues. To address this question, the authors took advantage of rat strains presenting differences in neural system excitability, selected by a quantifiable physiological trait - tibial nerve excitability threshold. Hence, they have collected bulk transcriptomic profiles of the hippocampus in high-excitability (LT) and low-excitability (HT) rats to characterize baseline inter-strain transcriptomic divergence.

Main comments

- The transcriptomic study fails to count with readouts of another part of the brain that could be considered as non-relevant for the excitability response, thus working as a negative control for potential differences among animals.

- Surprisingly, the differential gene expression analysis has been performed with a log2FoldChange filter >0.379…i.e. that the authors conserve/consider as differentially expressed genes those that have a fold-change of at least 1.3. This threshold appears quite mild. It would be interesting to know how many genes do pass a log2FoldChange filter >1; which is more classically used. Could such more stringent threshold still conserve the differences described by the authors?

-

- The discussion part contains several elements that should be retrieved in the results section.

Minor comments

- The presented figures are not ordered, and their quality are quite low.

- Figure 5&6 are missing color legends for edges and nodes.

Reviewer #2: The presented manuscript aims to study the inherited properties of the nervous system to respond to external environmental cues. To address this question, the authors took advantage of rat strains presenting differences in neural system excitability, selected by a quantifiable physiological trait - tibial nerve excitability threshold. Hence, they have collected bulk transcriptomic profiles of the hippocampus in high-excitability (LT) and low-excitability (HT) rats to characterize baseline inter-strain transcriptomic divergence.

Main comments

- The transcriptomic study fails to count with readouts of another part of the brain that could be considered as non-relevant for the excitability response, thus working as a negative control for potential differences among animals.

- Surprisingly, the differential gene expression analysis has been performed with a log2FoldChange filter >0.379…i.e. that the authors conserve/consider as differentially expressed genes those that have a fold-change of at least 1.3. This threshold appears quite mild. It would be interesting to know how many genes do pass a log2FoldChange filter >1; which is more classically used. Could such more stringent threshold still conserve the differences described by the authors?

-

- The discussion part contains several elements that should be retrieved in the results section.

Minor comments

- The presented figures are not ordered, and their quality are quite low.

- Figure 5&6 are missing color legends for edges and nodes.

Reviewer #3: This manuscript by Pavlova and colleagues presents a bulk RNA sequencing analysis of the hippocampus in rat strains selectively bred for contrasting levels of nervous system excitability, with the aim of characterising baseline interstrain transcriptomic divergence. The multi-layered analytical approach combines differential expression analysis, GO enrichment, and PPI network clustering. Several substantive concerns regarding animal welfare reporting, model characterisation, methodological transparency, and interpretive discipline must be addressed prior to publication.

Main Comments

1. PLOS ONE requires that manuscripts reporting animal research explicitly state, within the Methods section, the steps taken to ameliorate animal suffering. As currently written, the Methods do not adequately address this requirement. Specifically, the described euthanasia procedure — decapitation by guillotine without prior anaesthesia and in the absence of explicitly defined humane endpoints — appears difficult to reconcile with the requirements of the current European Directive 2010/63/EU, which mandates explicit humane endpoints and imposes stricter conditions on the justification of painful or distressing procedures than its predecessor, Directive 86/609/EEC.

2. The authors should expand their description and justification of the selective breeding model. A previous publication by Vylegzhanina and colleagues analysed rat strains with contrasting excitability levels across approximately 70 generations, whereas the present manuscript states that interstrain differences stabilised at the 10th generation, at which point between-strain variability exceeded within-strain variability by more than fourfold. This apparent discrepancy is not discussed, and the relationship between the earlier long-term breeding programme and the present cohort is unclear. The authors should explicitly clarify the generational history of the strains used here, discuss how phenotypic stability was monitored across generations, and situate the present cohort within the broader longitudinal context of the model. To this end, it would be highly valuable to include a figure or supplementary table illustrating the trajectory of excitability threshold divergence and within-strain variance across at least the first ten generations of selection, as this would provide readers with the empirical basis for the claim of phenotypic stabilisation and strengthen confidence in the genetic model underlying the transcriptomic comparisons.

3. The Results section would benefit from expansion and improved internal signposting. At present, the transition between analytical stages — from differential expression to GO enrichment to PPI network analysis — is abrupt, and summary statements foregrounding the key findings of each subsection are largely absent. Adding brief transitional paragraphs that synthesise the reasoning and highlight the principal outcomes of each analytical layer would substantially improve readability and logical flow. Furthermore, the RNA sequencing data generated in this study do not appear to have been deposited in a public repository. Deposition in an appropriate archive (e.g., NCBI GEO or ArrayExpress) prior to publication is strongly encouraged and is consistent with open-science standards increasingly required by journals in this field. The accession number should be reported in the Methods section.

4. The Discussion makes repeated cell-type-specific interpretive claims that are not supported by the bulk sequencing methodology employed. Attributing differences in Gfap expression to astrocytic reactivity, Slc2a5 to microglial metabolic state, P2rx7 to glial purinergic signalling, and Stab2 to an endothelial or myeloid neuroimmune profile implies a cellular resolution that bulk hippocampal RNA-seq fundamentally cannot provide. While the authors appropriately acknowledge in the Conclusion that their findings "motivate testable hypotheses to be evaluated in follow-up studies," this caveat is insufficiently foregrounded in the body of the Discussion, where speculative inferences frequently appear without explicit epistemic qualification. The authors are encouraged to systematically reframe cell-type-specific and mechanistic claims as hypotheses, accompanied in each case by a brief statement of what additional evidence — such as single-nucleus RNA sequencing, cell-type-resolved proteomics, or targeted immunohistochemistry — would be required to substantiate them.

5. The GO enrichment analysis is presented without adequate methodological detail or critical evaluation of its inherent limitations. The background gene set used is not reported, precluding reproducibility assessment. More substantively, terms such as "cognition," "learning and memory," and "regulation of synaptic plasticity" are among the most annotation-dense in GO databases and are prone to enrichment in virtually any transcriptomic dataset derived from neural tissue; their appearance cannot be interpreted as strong evidence of pathway-specific divergence without further qualification. The authors should report all enrichment parameters, apply a redundancy-reduction approach (e.g., semantic clustering via REVIGO or equivalent) to present a non-redundant summary of enriched terms, and explicitly distinguish high-specificity terms — which carry stronger inferential weight — from broad, annotation-rich terms that may reflect database structure as much as biology.

6. The protein-protein interaction network analysis using the STRING database with the discussion of individual cluster nodes — including SNAP25, NTRK2, MAPT, GFAP, AGT, FLT1, and PDGFRB — focuses primarily on their established biological roles as reported in the broader literature, rather than on their network-level properties (e.g., degree, betweenness centrality) or their specific quantitative contribution to interstrain transcriptional divergence. As a result, the PPI analysis functions largely as a thematically organised literature review rather than as an analytically independent layer of evidence. Connecting network topology explicitly to the magnitude of differential expression for each highlighted node would substantially strengthen the analytical value of this section.

6. PLOS authors have the option to publish the peer review history of their article (what does this mean?). If published, this will include your full peer review and any attached files.

Reviewer #1: No

Reviewer #2: No

Reviewer #3: No

---

## [Author Response · Author response to Decision Letter 1]

26 Apr 2026

Editor

We noticed you have some minor occurrence of overlapping text with the following previous publication(s), which needs to be addressed:

https://doi.org/10.1371/journal.pone.0323325

In your revision ensure you cite all your sources (including your own works), and quote or rephrase any duplicated text outside the methods section. Further consideration is dependent on these concerns being addressed.

We re-examined the revised manuscript for any duplicated wording outside the Methods section, including overlap with our own previously published work, and corrected the passages that could be considered too close in phrasing. Most of the overlap was limited to short, highly conventional technical or descriptive phrases, mainly in figure legends and brief transition sentences in the Results section. Wherever possible, we rephrased these passages to avoid unnecessary textual similarity.

4. Please note that funding information should not appear in any section or other areas of your manuscript. We will only publish funding information present in the Funding Statement section of the online submission form. Please remove any funding-related text from the manuscript.

Done

5. Please ensure that you refer to Figure 6 in your text as, if accepted, production will need this reference to link the reader to the figure.

Done

6. Please upload a new copy of Figures 5 and 6 as the detail is not clear. Please follow the link for more information: https://journals.plos.org/plosone/s/figures

Done

7. We are unable to open your Supporting Information file ( Supplementary_1 string_MCL_2_GO_1.tsv and Supplementary_2 string_MCL_2_GO_6.tsv) Please kindly revise as necessary and re-upload.

We have revised the Supporting Information files and converted the supplementary tables, including Supplementary_1 string_MCL_2_GO_1 and Supplementary_2 string_MCL_2_GO_6, into Excel to ensure they can be opened without technical difficulty.

The two reviewers raised several concerns about methodologies that are presented without enough details notably regarding animal welfare, description and justification of the selected model, and for the GO analysis. Furthermore, they found that PPI analysis does not reflect an independent layer of experimental evidence from the present study. Finally, it is not clear why the differential gene expression analysis has been performed with a log2FoldChange filter >0.379 and not >1 as classically used and how it may impact the conclusion. Although, I personally think that your manuscript has some merits, I invite you to address those important issues during revision.

We believe that these points have been addressed in the revised manuscript. In response to the reviewers’ comments, we clarified the animal welfare reporting, improved the description and justification of the selective breeding model, revised the GO analysis section for greater methodological transparency and interpretive caution, strengthened the PPI analysis by connecting network properties with differential expression metrics, and clarified the rationale for the selected log2FoldChange threshold.

Reviewer #1: Main comments

- The transcriptomic study fails to count with readouts of another part of the brain that could be considered as non-relevant for the excitability response, thus working as a negative control for potential differences among animals.

We agree that analysis of an additional brain region could help to assess regional specificity more fully. However, in this model, another brain area cannot be considered a true negative control, because inherited differences in nervous system excitability may affect multiple brain structures. At the same time, our previous transcriptomic study of the amygdala (Shalaginova, 2025 in References) showed a different pattern, with a much stronger immune-related signature, although the amygdala is also involved in stress-related regulation. This supports the view that the observed changes are not simply uniform brain-wide differences, but may vary across regions.

- Surprisingly, the differential gene expression analysis has been performed with a log2FoldChange filter >0.379…i.e. that the authors conserve/consider as differentially expressed genes those that have a fold-change of at least 1.3. This threshold appears quite mild. It would be interesting to know how many genes do pass a log2FoldChange filter >1; which is more classically used. Could such more stringent threshold still conserve the differences described by the authors?

-

We thank the reviewer for this important comment regarding the stringency of the criteria used to select differentially expressed genes when assessing inter-strain differences. We have now clarified that |log2FC| > 0.379 was not used as the primary criterion for DEG detection. Differential expression was defined primarily by FDR-adjusted significance (padj < 0.05), whereas the |log2FC| > 0.379 filter was applied only for visualization in the circular heatmap to reduce the number of displayed genes and highlight those with more pronounced between-strain differences.

The choice of a moderate fold-change threshold was based on the biological context of the study. The aim of our study was to compare the transcriptomic profiles of intact rats from selectively bred strains contrasting in nervous system excitability under normal physiological conditions. In this context, central regulatory mechanisms are expected to operate within the limits of homeostatic stability, supported by the evolutionary conservation of molecular and transcriptional processes in the brain. Therefore, large two-fold changes in gene expression are not necessarily expected, whereas moderate but statistically robust expression shifts may still be biologically meaningful. More stringent thresholds such as |log2FC| > 1 are useful for identifying large-amplitude transcriptional responses and may be particularly appropriate in models of severe brain injury, stress exposure or pathology.

We believe that using that using of this approach to the criterion for selecting DEGs allows us to retain genes with less pronounced but potentially biologically meaningful expression differences, as well as the pathways with which they are associated. This approach may provide insight into more subtle mechanisms regulating brain function under selective breeding for nervous system excitability and thereby enrich the interpretation of the results. Methodological studies in RNA-seq analysis have cautioned that stringent fold-change filtering may remove genes with genuine differential expression, particularly when expression changes are moderate but consistent across samples (Anqi Zhu, et al, Heavy-tailed prior distributions for sequence count data: removing the noise and preserving large differences, Bioinformatics, Volume 35, Issue 12, June 2019, Pages 2084–2092, https://doi.org/10.1093/bioinformatics/bty895).

To further support this rationale, we cite several neurotranscriptomic studies in which differential expression analyses were performed using low fold-change thresholds, including thresholds around 0.3 or without an explicit fold-change cutoff. This suggests that mild log2FC filtering is an accepted analytical option in neuroscience when the aim is to detect coordinated, moderate transcriptional differences rather than only large-magnitude expression changes.

• Goh JY, et al. Transcriptomic analysis of rat prefrontal cortex following chronic stress induced by social isolation - Relevance to psychiatric and neurodevelopmental illness, and implications for treatment. Neurobiol Stress. 2024 Oct 17;33:100679. doi: 10.1016/j.ynstr.2024.100679

• Chen MB, et al. Persistent transcriptional programmes are associated with remote memory. Nature. 2020 Nov;587(7834):437-442. doi: 10.1038/s41586-020-2905-5. Epub 2020 Nov 11. Erratum in: Nature. 2025 Aug;644(8077):E37. doi: 10.1038/s41586-025-09463-4

• Ayhan F, et al. Resolving cellular and molecular diversity along the hippocampal anterior-to-posterior axis in humans. Neuron. 2021 Jul 7;109(13):2091-2105.e6. doi: 10.1016/j.neuron.2021.05.003

• Mathys H, et al. Single-cell transcriptomic analysis of Alzheimer's disease. Nature. 2019 Jun;570(7761):332-337. doi: 10.1038/s41586-019-1195-2

• Makovka YV, et al. Effect of Short-Term Restraint Stress on the Expression of Genes Associated with the Response to Oxidative Stress in the Hypothalamus of Hypertensive ISIAH and Normotensive WAG Rats. Antioxidants (Basel). 2024 Oct 26;13(11):1302. doi: 10.3390/antiox13111302

As suggested by the reviewer, we additionally applied a more stringent |log2FC| > 1 threshold. This resulted in 38 genes, now listed in Supplementary Table 1. GO enrichment analysis of this restricted gene set produced only a small number of enriched terms, many of which were weakly related to nervous system function. Thus, applying a stringent |log2FC| > 1 threshold would address a different question: which genes show large-amplitude expression changes. Whereas our study aimed to characterize the functional organization of moderate but coordinated inter-strain differences in gene expression.

The discussion part contains several elements that should be retrieved in the results section.

We revised the manuscript to improve the separation between the Results and Discussion sections. Specifically, we moved or condensed several result-like summary elements that had previously appeared in the Discussion, especially where the text described functional groupings of DEGs, enriched GO categories, or the structure of PPI clusters.

Minor comments

- The presented figures are not ordered, and their quality are quite low. Fixed

- Figure 5&6 are missing color legends for edges and nodes. Fixed

Reviewer #2: Main comments

1. PLOS ONE requires that manuscripts reporting animal research explicitly state, within the Methods section, the steps taken to ameliorate animal suffering. As currently written, the Methods do not adequately address this requirement. Specifically, the described euthanasia procedure — decapitation by guillotine without prior anaesthesia and in the absence of explicitly defined humane endpoints — appears difficult to reconcile with the requirements of the current European Directive 2010/63/EU, which mandates explicit humane endpoints and imposes stricter conditions on the justification of painful or distressing procedures than its predecessor, Directive 86/609/EEC.

We re-examined this point in light of the current European Directive 2010/63/EU. We note that the Directive requires that methods of euthanasia should minimize pain, suffering, and distress, and that the appropriate species-specific method should be used. Also, decapitation in rodents is treated as a method to be used when other approaches are not suitable in the specific experimental context. In our study, the terminal procedure was selected on the basis of harm-benefit balance and endpoint validity. The primary outcome was transcriptomic analysis of immune- and cytokine-related signals in intact hippocampal tissue. Inhalational anesthesia such as isoflurane can rapidly alter pro-inflammatory gene expression and therefore would directly compromise the main molecular readout of the study. Under these conditions, the use of prior anesthesia would reduce the scientific validity of the terminal sampling procedure and could lead to animal use without obtaining interpretable data. We have clarified this rationale in the revised Methods and have expanded the description of measures taken to minimize suffering, including routine welfare monitoring, minimization of handling time, and rapid performance of the terminal procedure by experienced personnel.

2. The authors should expand their description and justification of the selective breeding model. A previous publication by Vylegzhanina and colleagues analysed rat strains with contrasting excitability levels across approximately 70 generations, whereas the present manuscript states that interstrain differences stabilised at the 10th generation, at which point between-strain variability exceeded within-strain variability by more than fourfold. This apparent discrepancy is not discussed, and the relationship between the earlier long-term breeding programme and the present cohort is unclear. The authors should explicitly clarify the generational history of the strains used here, discuss how phenotypic stability was monitored across generations, and situate the present cohort within the broader longitudinal context of the model. To this end, it would be highly valuable to include a figure or supplementary table illustrating the trajectory of excitability threshold divergence and within-strain variance across at least the first ten generations of selection, as this would provide readers with the empirical basis for the claim of phenotypic stabilisation and strengthen confidence in the genetic model underlying the transcriptomic comparisons.

Thank you for this important comment. We agree that the wording in the original version was not sufficiently clear and could create confusion regarding the generational history of the model. The statement that interstrain differences had stabilized by the 10th generation refers to the early phase of the selective breeding program, that is, the point at which divergence between the strains became clearly established and exceeded within-strain variability by more than fourfold. It does not refer to the generation used in the present experiment. The animals studied in the current study belonged to the 76th generation of the long-term breeding program.

We have clarified this explicitly in the revised manuscript. We have also added text explaining that nervous system excitability was monitored both during the selective breeding process, when animals were chosen for further breeding, and during the formation of experimental groups. In addition, we have included supplementary material showing individual excitability thresholds, to better place the present cohort within the broader context of the model (Supplementary Figure 1).

3. The Results section would benefit from expansion and improved internal signposting. At present, the transition between analytical stages — from differential expression to GO enrichment to PPI network analysis — is abrupt, and summary statements foregrounding the key findings of each subsection are largely absent. Adding brief transitional paragraphs that synthesise the reasoning and highlight the principal outcomes of each analytical layer would substantially improve readability and logical flow.

In the revised manuscript, we added brief transitional and summary sentences to improve the logical flow between differential expression analysis, functional enrichment, and PPI analysis.

Furthermore, the RNA sequencing data generated in this study do not appear to have been deposited in a public repository. Deposition in an appropriate archive (e.g., NCBI GEO or ArrayExpress) prior to publication is strongly encouraged and is consistent with open-science standards increasingly required by journals in this field. The accession number should be reported in the Methods section.

We deposited the raw and processed data in NCBI and added the corresponding link in the Methods section. (GSE327807).

4. The Discussion makes repeated cell-type-specific interpretive claims that are not supported by the bulk sequencing methodology employed. Attributing differences in Gfap expression to astrocytic reactivity, Slc2a5 to microglial metabolic state, P2rx7 to glial purinergic signalling, and Stab2 to an endothelial or myeloid neuroimmune profile implies a cellular resolution that bulk hippocampal RNA-seq fundamentally cannot provide. While the authors appropriately acknowledge in the Conclusion that their findings "motivate testable hypotheses to be evaluated in follow-up studies," this caveat is insufficiently foregrounded in the body of the Discussion, where speculative inferences frequently appear without explicit epistemic qualification. The authors are encouraged to systematically reframe cell-type-specific and mechanistic claims as hypotheses, accompanied in each case by a brief statement of what additional evidence — such as single-nucleus RNA sequencing, cell-type-resolved proteomics

---

## [Decision Letter · Decision Letter 1]

13 May 2026

PONE-D-26-04533R1Transcriptomic profile of the hippocampus of rat strains with contrasting nervous system excitabilityPLOS One

Dear Dr. Shalaginova,

Thank you for submitting your manuscript to PLOS ONE. After careful consideration, we feel that it has merit but does not fully meet PLOS ONE’s publication criteria as it currently stands. Therefore, we invite you to submit a revised version of the manuscript that addresses the points raised during the review process.

We look forward to receiving your revised manuscript.

Kind regards,

Jean-Pierre Mothet, Ph.D

Academic Editor

PLOS One

Journal Requirements:

**Additional Editor Comments:**

The reviewer and I appreciate your efforts in addressing adequately the original concerns. We particularly acknowledge the integration of new figures, tables and detailed revised sections. Accordingly, the revised version of your manuscript has been largely improved. However, there are still few minor comments raised by the reviewer that require your attention. Therefore, I kindly ask you to address these remaining comments before final acceptance.

Reviewers' comments:

Reviewer's Responses to Questions

**Comments to the Author**

1. If the authors have adequately addressed your comments raised in a previous round of review and you feel that this manuscript is now acceptable for publication, you may indicate that here to bypass the “Comments to the Author” section, enter your conflict of interest statement in the “Confidential to Editor” section, and submit your "Accept" recommendation.

Reviewer #3: (No Response)

2. Is the manuscript technically sound, and do the data support the conclusions?

Reviewer #3: Yes

3. Has the statistical analysis been performed appropriately and rigorously? 

Reviewer #3: Yes

4. Have the authors made all data underlying the findings in their manuscript fully available?

Reviewer #3: Yes

5. Is the manuscript presented in an intelligible fashion and written in standard English?

Reviewer #3: Yes

6. Review Comments to the Author

Reviewer #3: The authors have largely addressed my previous comments satisfactorily. The deposition of RNA sequencing data in the NCBI GEO public repository under accession number GSE327807 is particularly appreciated. The revised manuscript, which now includes additional figures, tables, and revised sections, is in my view near ready for acceptance. I have a few follow-up remarks and comments that I would ask the authors to address before final acceptance.

1. PCA analysis

The current presentation of the PCA based solely on the 654 differentially expressed genes (DEGs) is methodologically problematic, as it introduces a circular analysis issue — also known as "double dipping": since genes were selected precisely because they differ between groups, the resulting separation and the variance explained by the first principal components are artificially inflated. Importantly, a PCA performed on all expressed and variable genes used as background for the DESeq2 analysis already achieves a clear and complete separation of the HT and LT groups into non-overlapping clusters, which constitutes a far more rigorous and compelling demonstration of transcriptomic divergence between the two conditions. The authors should therefore present the PCA based on all background genes as the primary analysis, and may retain the DEG-based PCA as a supplementary figure for comparison, clearly contrasting the variance explained in both cases and explicitly discussing the methodological distinction.

2. Protein-protein interaction (PPI) enrichment analysis

The significant PPI enrichments reported for networks derived from GO biological process gene sets are fully expected by construction, as genes sharing a biological process annotation are inherently more likely to interact, and therefore these results add limited interpretive value to the manuscript. A more informative and meaningful observation would be to highlight that the full set of 654 DEGs (FDR < 0.05) itself shows significant PPI enrichment, which provides evidence that the transcriptional response to the experimental contrast is not a collection of independent gene-level changes, but rather reflects the perturbation of coordinated biological networks.

STRING network:

number of nodes: 639

number of edges: 547

expected number of edges: 381

PPI enrichment p-value: 7.77e-16

https://version-12-0.string-db.org/cgi/network?networkId=bsNRxR3QJrRd

This PPI enrichment should be presented as a key result, as it both strengthens the biological relevance of the findings and directly justifies the more focused PPI analyses conducted on the DEGs associated with the GO biological processes "regulation of the MAPK cascade" and "regulation of synaptic plasticity."

7. PLOS authors have the option to publish the peer review history of their article (what does this mean?). If published, this will include your full peer review and any attached files.

Reviewer #3: No

---

## [Author Response · Author response to Decision Letter 2]

15 May 2026

Reviewer #3:

1. PCA analysis

The current presentation of the PCA based solely on the 654 differentially expressed genes (DEGs) is methodologically problematic, as it introduces a circular analysis issue — also known as "double dipping": since genes were selected precisely because they differ between groups, the resulting separation and the variance explained by the first principal components are artificially inflated. Importantly, a PCA performed on all expressed and variable genes used as background for the DESeq2 analysis already achieves a clear and complete separation of the HT and LT groups into non-overlapping clusters, which constitutes a far more rigorous and compelling demonstration of transcriptomic divergence between the two conditions. The authors should therefore present the PCA based on all background genes as the primary analysis, and may retain the DEG-based PCA as a supplementary figure for comparison, clearly contrasting the variance explained in both cases and explicitly discussing the methodological distinction.

We thank the reviewer for this important methodological comment. We performed an additional PCA using the transcriptome-wide background expression matrix without DEG-based gene selection. All expressed genes were included after removal of genes with zero expression or zero variance across samples. The revised manuscript now presents both the DEG-based PCA (Fig. 2A) and the background-gene PCA (Fig. 2B). Importantly, clear separation between LT and HT samples was also preserved in the unbiased background PCA, although the variance explained by the first principal components was lower, as expected for a transcriptome-wide analysis not restricted to DEGs. The Methods and Results sections were revised accordingly.

2. Protein-protein interaction (PPI) enrichment analysis

The significant PPI enrichments reported for networks derived from GO biological process gene sets are fully expected by construction, as genes sharing a biological process annotation are inherently more likely to interact, and therefore these results add limited interpretive value to the manuscript. A more informative and meaningful observation would be to highlight that the full set of 654 DEGs (FDR < 0.05) itself shows significant PPI enrichment, which provides evidence that the transcriptional response to the experimental contrast is not a collection of independent gene-level changes, but rather reflects the perturbation of coordinated biological networks.

STRING network:

number of nodes: 639

number of edges: 547

expected number of edges: 381

PPI enrichment p-value: 7.77e-16

https://version-12-0.string-db.org/cgi/network?networkId=bsNRxR3QJrRd

This PPI enrichment should be presented as a key result, as it both strengthens the biological relevance of the findings and directly justifies the more focused PPI analyses conducted on the DEGs associated with the GO biological processes "regulation of the MAPK cascade" and "regulation of synaptic plasticity."

We thank the reviewer for this helpful suggestion and analysis. The Results section was revised to include this global PPI enrichment before the focused PPI analyses. The Discussion was also modified to interpret this result in the context of inherited nervous system excitability as a polygenic trait. The global STRING network was added as Supplementary Fig. 2 and node degrees as Supplementary Table 8.

---

## [Editor Report · Decision Letter 2]

18 May 2026

Transcriptomic profile of the hippocampus of rat strains with contrasting nervous system excitability

PONE-D-26-04533R2

Dear Dr. Shalaginova,

We’re pleased to inform you that your manuscript has been judged scientifically suitable for publication and will be formally accepted for publication once it meets all outstanding technical requirements.

Kind regards,

Jean-Pierre Mothet, Ph.D

Academic Editor

PLOS One

Additional Editor Comments (optional):

The authors have satisfactorily addressed all concerns

---

## [Editor Report · Acceptance letter]

PONE-D-26-04533R2

PLOS One

Dear Dr. Shalaginova,

I'm pleased to inform you that your manuscript has been deemed suitable for publication in PLOS One. Congratulations! Your manuscript is now being handed over to our production team.

Kind regards,

on behalf of

Dr Jean-Pierre Mothet

Academic Editor

PLOS One